# The effect of different anesthetics on the incidence of AKI and AKD after neurosurgical procedures

Vedran Premuzic[1,2]*, Vasilije Stambolija[3], Marin Lozic[3], Josip Kovacevic[3], Vladimir Prelevic[4], Marina Peklic[1], Miroslav Scap[3], Ante Sekulic[2,3], Nikolina Basic-Jukic[1,2], Slobodan Mihaljevic[2,3], Kianoush B. Kashani[5,6]

1 Department of Nephrology, Hypertension, Dialysis and Transplantation, University Hospital Center Zagreb, Zagreb, Croatia, 2 School of Medicine, University of Zagreb, Zagreb, Croatia, 3 Clinic of Anesthesiology Resuscitation and Intensive Care, University Hospital Center Zagreb, Zagreb, Croatia, 4 Department of Nephrology and Dialysis, Clinical Center Podgorica, Podgorica, Montenegro, 5 Department of Medicine, Division of Nephrology and Hypertension, Division of Pulmonary and Critical Care Medicine, Mayo Clinic, Rochester, NY, United States of America, 6 Department of Medicine, Division of Nephrology and Hypertension, Mayo Clinic, Rochester, MN, United States of America

* vpremuzic@gmail.com

**Data Availability Statement:** Data cannot be shared publicly because of the local IRB [UHC Zagreb IRB] policies. Data are available from the UHC Zagreb IRB Institutional Data Access / Ethics

## Abstract

Acute kidney injury (AKI) incidence after neurosurgical operations has been reported as 10–14%. The literature regarding the incidence of nosocomial acute kidney disease (AKD) following neurosurgery is scarce. This retrospective, single-center, observational study aimed to assess the impact of different anaesthetics on development of postoperative AKI and persistent AKD in neurosurgical patients. We have categorized patients depending by the type of total intravenous anaestesia with propofol or sevoflurane. Most patients (74%) were on total intravenous anesthesia with propofol, while the rest (26%) were on sevoflurane. Patients were divided into subgroups with and without AKD depending on glomerular filtration rate <or> 60 ml/min regarding kidney function at the end of intensive care unit stay. AKI was diagnosed in 341 (5.39%) patients. Significantly higher number of patients developed AKD in the sevoflurane group (16.9% vs. 6.3%). There was a significantly higher number of patients with both high and low AKI stages on sevoflurane and with hypotension during operation. Anaesthesia with sevoflurane had increased OR of 5.09 and ROC value of 0.681 for development of AKI. Anesthesia with sevoflurane had an increased OR of 4.98 and ROC value of 0.781 for development of AKD. Mortality was independently associated with anesthesia with sevoflurane, AKI development, hypotension during operation and AKD. Anesthesia with sevoflurane, hypotension during operation, and the development of AKD at the end of ICU stay were associated with higher mortality in the whole group (HR 6.996, HR 1.924 and HR 4.969, respectively). Patients treated with balanced anesthesia with sevoflurane had more frequent AKI and AKD with shorter survival. Renal toxicity of sevoflurane is pronounced in hypotension during operation and with a history of diabetes and coronary disease.

Committee (contact via kbc-zagreb@kbc-zagreb.hr) for researchers who meet the criteria for access to confidential data. The data underlying the results presented in the study are available from UHC Zagreb IRB (contact information:kbc-zagreb@kbc-zagreb.hr). The data set (an excel database) is available for every researcher interested in accessing the data regarding this study after contacting the local IRB by email (kbc-zagreb@kbc-zagreb.hr). The researchers need to cite the study approval number (02/21-JG) and state that after accessing the data they will only used them for scientific work. The authors of this study did not received any special privileges in accessing the data that other researchers would not have.

**Funding:** The author(s) received no specific funding for this work.

**Competing interests:** The authors have declared that no competing interests exist.

## Introduction

Postoperative acute kidney injury (AKI) is common. It is associated with significant morbidity, including chronic kidney disease (CKD) and end-stage kidney disease (ESKD), prolonged intensive care unit (ICU) and hospital stay, and higher economic burden [1–4]. AKI is also associated with increased mortality even during modest kidney injury and independent of other risk factors [5,6].

Incidence of AKI ranges between 20 to 67% in medical and surgical ICU patients, being lower in elective surgery patients and highest in septic patients [1,7,8]. Studies of AKI after neurosurgical operations showed an incidence of 10–14% [9,10].

The independent risk factors for postoperative AKI in neurosurgical patients include intraoperative blood loss, need for reoperation, dehydration or osmotic nephrosis following intraoperative mannitol administration, and postoperative Acute Physiology and Chronic Health Evaluation II (APACHE II) score [10]. In addition, anesthetic procedures and the selection of the anesthetic are also related to the development of AKI–a meta-analysis showed that the incidence of AKI is lower for neuraxial anesthesia when compared with general anesthesia [11]. For example, propofol is associated with reduced incidence and severity of AKI in contrary to sevoflurane [12]. AKD is a recently recognized entity that is caused by kidney injury. AKD describes acute or subacute damage and/or loss of kidney function for a duration of between 7 and 90 days after exposure to an AKI-initiating event [13]. The development of AKD indicates changing renal function that could be strongly associated with long-term renal outcomes and major adverse kidney events (MAKEs) leading to CKD progression, ESKD, and mortality [14]. The risk of developing CKD/ESRD after AKI increases in a graded fashion dependent on the severity of AKI (pooled adjusted hazard ratio (HR) for CKD 8.8, 95% CI 3.1–25.5; pooled adjusted HR for ESRD 3.1, 95% CI 1.9–5.0) [15].

Therefore, this study aimed to assess the impact of different anaesthetics on development of postoperative AKI and persistent acute kidney disease (AKD) in neurosurgical patients.

## Materials and methods

Consecutive 6,317 adults ($\geq$18 years old) who were addmited for neurosurgical procedures between January 2010 and December 2019 was enrolled in this retrospective, single-center, observational study. The study was approved by the hospital ethics committee (UHC Zagreb, Croatia, IRB approval number:02/21-JG; approval date 14.04.2021.; Study title: Extracorporeal blood purification and continuous renal replacement therapy in critical patient and/or in patients with COVID-19 infection) and have been performed in accordance with the ethical standards as laid down in the 1964 Declaration of Helsinki and its later amendments or comparable standards. Data were handled in agreement with patient informed consent. Patients were included if they were $\geq$18 years old and admitted to the hospital for neurosurgical procedures. Patients with a history of CKD at any stage and those with missing data during the ICU stay were excluded. CKD was defined as kidney damage or glomerular filtration rate (GFR) <60 mL/min/1.73 m2 for at least three months, irrespective of cause, and was adjudicated by reviewing medical records [16]. The baseline creatinine was measured rutinely two weeks before admission. We have considered at least one postoperative creatinine values as inclusion criterium. We have categorized patients depending by the type of operative procedure, 1): anurysmal and arteriovenous vascular malformation vascular surgery, 2) surgery for subdural haematoma, and 3) craniotomy for tumour. Surgical data, including duration of surgery, hypotension during operation, and type of anaesthesia, were recorded. The anaestetic procedure was standardized. Anaesthesia was induced with propofol (2.0 mg/kg or to the desired effect), opioids (0.1–0.3 ug/kg of sufentanil citrate), and rocuronium bromide was used as a muscle

relaxant (0.6 mg/kg), and maintained with total intravenous anaesthesia with propofol (0.1–0.2 mg/kg/min) or balanced type of anaesthesia with sevoflurane (minimal alveolar concentration (MAC) dose for 25 years old patients was 2.6 vol%, for 40 years old patients was 2.1 vol% and for 60 years old patients was 1.7 vol%) in combination with opioids (0.3–0.5 ug/kg/h of sufentanil citrate) and bolus of rocuronium bromide of 0.1 mg/kg. The reasons for choosing different anesthetics were mostly based on the type of surgery. Propofol was used for patients with aneurysmal and arteriovenous vascular malformation, vascular surgery, and craniotomy for tumors. In contrast, sevoflurane was used for subdural hematoma or for urgent operations. Sevoflurane was administered with fresh gas flow rates at least 2 L/min for exposures greater than 1 hour as well as propofol. All subjects received paracetamol 2 g p.o. 1 h preoperatively. Blood loss was replaced with albumin 5% in a 1:1 ratio until haemoglobin became <70 g/L, when transfusion with blood products was initiated. Intraoperative hypotension was defined as systolic blood pressure below 80 mmHg or a decrease in systolic blood pressure of more than 20% below baseline, or systolic blood pressure below 100 mmHg and/or 30% decrease below the baseline.

RIFLE criteria were used for the diagnosis of AKI by comparing changes in serum creatinine levels during hospitalization to baseline levels before admission to the hospital [17]. Patients with AKI were stratified according to the severity or the need for renal replacement therapy. Patients who were treated with renal replacement therapy (failure) in the first two days after the surgical procedure were defined as high-stage AKI, while patients with a 150% increase in serum creatinine and urinary output<0.5ml/kg/h lasting 6 hours (risk) and 200% increase in serum creatinine and urinary output<0.5ml/kg/h lasting 12 hours (injury) were defined as low stage AKI. All continuous renal replacement therapy (CRRT) procedures were performed with the mean blood flow rate between 200 mL/min and 250 mL/min, depending on the blood-access function and desired ultrafiltration rates. We aimed at the dose of dialysis >35 ml/kg/hour. Baseline and follow-up renal function was defined as an estimated GFR (mL/min), which was estimated daily using the Chronic Kidney Disease-Modification of Diet in Renal Disease (CKD-MDRD) equation. Patients were divided into two subgroups regarding AKD at the end of ICU stay; the first AKD group with GFR<60 ml/min and the second without AKD group with GFR≥60 ml/min. Renal outcome at the time of discharge was evaluated by comparing the end of ICU stay serum creatinine levels with the baseline serum creatinine levels.

Patients' clinical and laboratory data were collected from electronic hospital databases. The collected variables included age, sex, body mass index (BMI), preexisting clinical conditions like diabetes, arterial hypertension, coronary disease, and the American Society of Anesthesiologists (ASA) classification. In addition, the postoperative Glasgow coma scales were assessed immediately after patients had recovered from the anesthesia. Data were collected before and after the operation and at least once daily as a part of routine clinical care during ICU hospitalization. Follow-up lasted until the last enrolled patient reached the 730-day or death from complications of the primary disease or cardiovascular event. The Croatian National Public Health Institute records were used for the patient survival status after discharge.

## Statistical analysis

Normality of data distribution was tested using the Kolmogorov-Smirnov test. Preliminary analyses were performed to ensure no violation of normality, linearity, and homoscedasticity assumptions. Categorical data were expressed as numbers and frequencies. Correlations were obtained using Pearson's test for normally distributed variables and Spearman rank correlation for non-normally distributed variables. Normally distributed variables were presented as

means ± standard deviations, and Student's t-test for independent samples was used to compare the two groups. Non-normally distributed data were presented as median and interquartile ranges, and Mann-Whitney

U-test was used in the comparison between the two groups. Analysis of variance (ANOVA) was used to detect significant differences among ≥ two groups. Categorical variables were compared using the Chi-square test. Survival analysis was done with Kaplan-Meier curves tested with the log-rank test, while hazard ratios were estimated with Cox proportional hazards regression. Multiple linear regression was used to explore the influence of different variables on serum creatinine levels at discharge from ICU, while logistic regression was used for categorical dependent variables. The discriminant ability was assessed with Receiver Operating Characteristic (ROC) analysis. Statistical analysis was performed using SPSS version 23.0 (IBM Corp., USA). A p-value <0.05 (two-sided tests) was considered significant.

## Results

Among 13,165 patients who were screened, 6,317 patients (4,477 men, 1,840 women; average age 57.9+3.3) entered the final analysis (**Fig 1**).

The procedure type distribution was 3,181 (50.3%) for aneurism, 1,289 (20.4%) for brain tumors, and 1,847 (29.3%) for hematoma. Most patients (76.3%) were on total intravenous anesthesia with propofol, while the rest (23.7%) were on sevoflurane. The median length of ICU stay for all patients, i.e., the propofol and the sevoflurane groups, were 24 (12–36) days. There were 1138 (18%) patients with prior arterial hypertension, 567 (8.9%) with prior diabetes, and 409 (6.4%) with prior coronary disease.

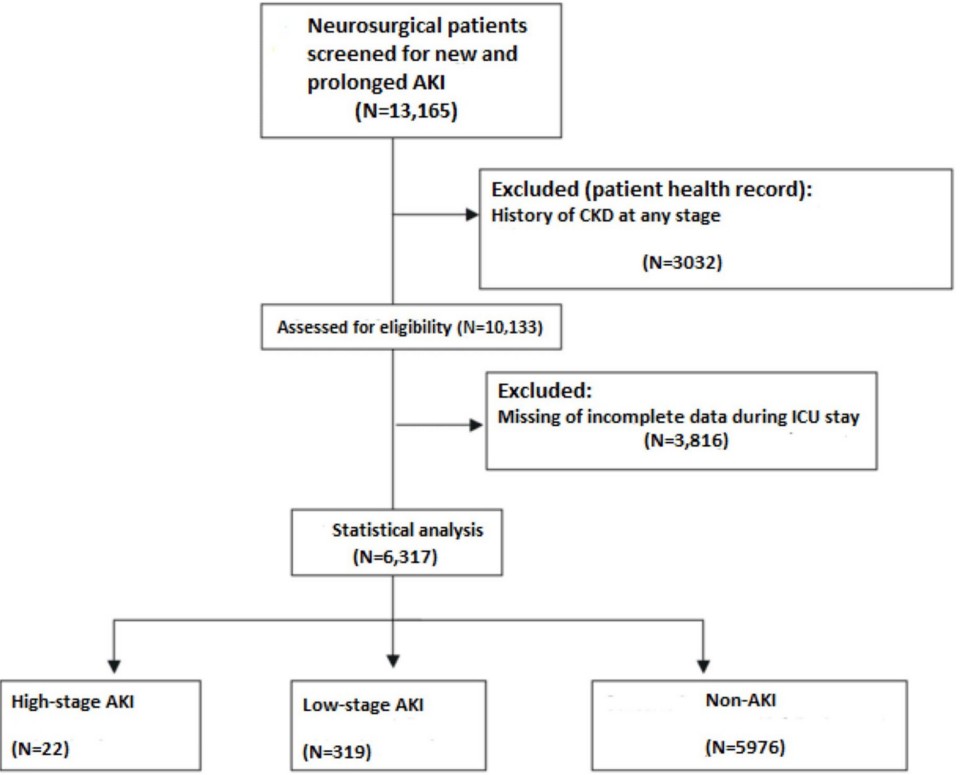

**Fig 1. A patient flowchart.** ICU-intensive care unit; AKI-acute kidney injury.

Demographic parameters, comorbidities, clinical variables, and laboratory parameters between patients on sevoflurane and propofol are demonstrated in **Table 1**.

We did not find differences in the number of patients who developed AKI, but there was a significantly higher number of patients who developed AKD in the sevoflurane group (16.9% vs. 6.3%). We did not find differences in age, sex, BMI, comorbidities, intraoperative parameters, or in administered doses of anesthetics, muscle relaxants, and opioids during anesthesia induction and during operation between these subgroups of patients. In logistic regression, we did not find a significant association between different variables for the development of AKI.

## Variables associated with AKI development

Demographic parameters, comorbidities, clinical variables, and laboratory parameters between patients with high and low stages of AKI and patients without AKI after the neurosurgical procedure are demonstrated in **Table 2**.

AKI was diagnosed in 341 (5.39%) patients and AKD was diagnosed in 562 (8.89%) patients. We did not find differences in age, sex, BMI, indications for neurosurgery, intraoperative parameters or in administered doses of anaesthetics, muscle relaxants and opioids during anaesthesia induction and during operation between these subgroups of patients except higher MAC doses in the group with high stage of AKI. There was a significantly higher number of patients on sevoflurane and patients with diabetes and coronary disease in groups with high and low stage of AKI compared to patients without AKI. We found a significantly higher number of patients with hypotension during operation in both high and low AKI grades than individuals without AKI. On logistic regression anaesthesia with sevoflurane had increased OR of 5.09 [CI 0.02,0.77] while hypotension during operation has almost significant association and increased OR of 3.43 [CI 0.87, 1.80] for development of AKI (**Table 3**). Interestingly, we haven´t found association of age and comorbidities like diabetes and coronary disease with development of AKI.

ROC analysis was computed to test the discriminative power of the administration of sevoflurane on AKI development. ROC value was 0.681 [95% CI: 0.572–0.789], $p < 0.001$ (**Fig 2A**).

## Variables associated with AKD development at the time of dismissal from ICU

Demographic parameters, comorbidities, clinical variables, and laboratory parameters between patients with AKD and without AKD at the end of ICU stay are demonstrated in **Table 4**.

A significantly higher number of patients with AKD were on anesthesia with sevoflurane and with hypotension during operation that patients without AKD. We did not find differences in age, sex, BMI, indications for neurosurgery or in administered doses of anaesthetics, muscle relaxants and opioids during anaesthesia induction and during operation between these subgroups of patients. As expected, significantly higher number of patients developed AKI and had lower GFR levels after the neurosurgical procedure in AKD group. Also, AKD patients had significanty lower values of mean blood pressure values after neurosurgical procedures. There was a significantly higher number of patients with diabetes and coronary disease in the AKD group. In the linear regression model, balanced type of anaesthetic, development of AKI, and hypotension during the operation were predictors for higher serum creatinine levels and, consequently, development of AKD at the end of ICU stay (β = 0.558, p = 0.028;β = 0.612, p = 0.013;β = 0.405, p = 0.043). Interestingly, age and comorbidities like diabetes and coronary disease were not associated with higher creatinine levels. In another linear regression analysis model, we analyzed the possible interaction between the use of sevoflurane and

**Table 1. Demographic parameters, comorbidities, and differences in clinical variables and laboratory parameters between patients on sevoflurane and propofol.**

| Variables | Sevoflurane (N = 1501) | Propofol (N = 4816) | p |
|---|---|---|---|
| Age (years) | 53.±3.0 | 57.1±3.2 | 0.13 |
| Sex (males) N(%) | 985 (65.6) | 3450 (71.6) | 0.32 |
| BMI | 27.1±2.2 | 27.2±2.3 | 0.89 |
| Comorbidities N(%) *Diabetes* *Hypertension* *Coronary heart disease* | 151 (10.0) 272 (18.1) 83 (5.5) | 416 (8.6) 866 (17.9) 326 (6.7) | 0.46 0.63 0.58 |
| Indication for surgery *Aneurism* *Tumour* *Subdural haematoma* | 182 (12.2) 32 (2.1) 1287 (85.7) | 2189 (45.4) 2493 (51.8) 134 (2.8) | <0.001 <0.001 <0.001 |
| ASA score | 2.7±0.8 | 2.70±0.6 | 0.55 |
| GCS score | 10.±1.2 | 11.8±1.8 | 0.23 |
| Duration of surgery (minutes) | 299.8±25.8 | 347.1±32.5 | 0.12 |
| Anaesthesia induction *Propofol dose (mg/kg)* *Opioid dose (mg/kg)* *Muscle relaxant dose (mg/kg)* Intravenous anaesthesia *Propofol dose (mg/kg/min)* *Opioid dose (mg(kg/h)* *Muscle relaxant dose (bolus mg/kg)* | 2.1±0.5 0.22 (0.10–0.31) 0.5±0.1 0.12 (0.10–0.16) 0.25 (0.12–0.33) 0.11 (0.09–0.13 | 2.2±0.5 0.17 (0.10–0.26) 0.61±0.1 0.11 (0.10–0.14) 0.22 (0.11–0.31) 0.10 (0.08–0.12) | 0.87 0.34 0.56 0.36 0.69 0.51 |
| AKI Yes N(%) AKD Yes N(%) | 138 (9.1) 255 (16.9) | 185 (3.8) 307 (6.3) | 0.18 <0.001 |
| Hemodynamic status | | | |
| Surgery: *Hypotension during operation Yes N(%)* *Mean arterial pressure (mmHg)* *Mean administration of fluids (mL/h)* *Hemorrhage (mL)* | 172 (11.4) 64.2±8.9 620±43.7 145±18.1 | 645 (13.4) 63.3±8.2 595±41.2 155±18.8 | 0.28 0.66 0.43 0.19 |
| Systolic blood pressure (mmHg) baseline | 147.6±17.0 | 145.1±16.8 | 0.78 |
| Systolic blood pressure (mmHg) after | 134.4±16.1 | 140.2±16.3 | 0.39 |
| Diastolic blood pressure (mmHg) baseline | 66.7±9.8 | 70.7±10.2 | 0.51 |
| Diastolic blood pressure (mmHg) after | 65.4±9.5 | 71.6±10.1 | 0.16 |
| Mean arterial pressure (mmHg) baseline | 81.±10.4 | 82.3±10.5 | 0.72 |
| Mean arterial pressure (mmHg) after | 63.8±8.4 | 64.1.2±9.5 | 0.84 |
| CVP (mmHg) baseline | 12 (9–13) | 12 (8–13) | 0.66 |
| CVP (mmHg) after | 7 (5–10) | 9 (6–11) | 0.22 |
| Urinary output (ml/h) after | 75 (44–102) | 82 (51–113) | 0.31 |
| Laboratory data | | | |
| Sodium (mmol/l) baseline | 137 (134–140) | 136 (133–138) | 0.59 |
| Sodium (mmol/l) after | 137 (134–141) | 135 (131–137) | 0.18 |
| WBC (x10$^9$/l) baseline | 10.6±1.7 | 13.±1.9 | 0.14 |
| WBC (x10$^9$/l) after | 9.2±0.4 | 12.8±2.1 | 0.03 |
| Platelets (x10$^9$/l) baseline | 180 (151–202) | 176 (151–199) | 0.55 |
| Platelets (x10$^9$/l) after | 178 (142–201) | 187 (161–211) | 0.38 |
| Creatinine (umol/l) baseline | 88.2±5.2 | 82.4±3.9 | 0.43 |
| Creatinine (umol/l) after | 90.4±6.1 | 95.1±6.4 | 0.38 |
| GFR (ml/min/1.73m$^2$) baseline | 90 (55–129) | 96 (62–133) | 0.22 |
| GFR (ml/min/1.73m$^2$) after | 88 (7–18) | 81 (28–77) | 0.11 |
| BUN (umol/l) baseline | 9.2±1.5 | 8.8±1.4 | 0.45 |
| BUN (umol/l) after | 9.±1.9 | 10.2±2.4 | 0.67 |

*(Continued)*

**Table 1.** (Continued)

| Variables | Sevoflurane (N = 1501) | Propofol (N = 4816) | p |
|---|---|---|---|
| C-reactive protein (mg/l) baseline | 12 (6–20) | 56 (28–74) | 0.02 |
| C-reactive protein (mg/l) after | 7 (3–18) | 57 (29–76) | <0.01 |
| Survival days | 433 (332–561) | 498 (361–624) | 0.30 |

AKD-acute kidney disease; AKI-acute kidney injury; BMI-body mass index; ASA-American Society of Anesthesiologists; GCS-Glasgow coma score; ICU-intensive care unit; CVP-central venous pressure; WBC-white blood count; GFR-estimated glomerular filtration rate; BUN-blood urea nitrogen; results are shown as mean +/- SD or median (interquartile range).

hypotension during the operation and the development of AKI and AKD. We did not find any statistically significant interaction for these two variables (β = 0.487, p = 0.602;β = 0.522, p = 0.490). On logistic regression, anesthesia with sevoflurane had an increased OR of 4.98 [CI 0.79, 1.15] for AKD development (**Table 3**). We did not note associations of age and comorbidities like diabetes and coronary disease with AKD development. ROC analysis was computed to test the discriminative power of sevoflurane administration on the development of AKD. ROC values was 0.781 [95% CI: 0.668–0.904], p < 0.001 (**Fig 2B**).

## Survival of patients with development of AKI and AKD

In the linear regression analysis in the entire cohort, mortality was independently associated with anesthesia with sevoflurane (β = 0.412, p = 0.041), AKI development (β = 0.532, p = 0.033), hypotension during operation (β = 0.488, p = 0.039) and AKD (β = 0.556, p = 0.022). In another linear regression analysis model, we analyzed the possible interaction between the use of sevoflurane and hypotension during the operation and mortality. We did not find any statistically significant interaction for these variables (β = 0.329, p = 0.572;β = 0.288, p = 0.413). Anesthesia with sevoflurane, hypotension during operation, and the development of AKD at the end of ICU stay were associated with higher mortality in the whole group (HR 6.996 [1.04, 3.88], HR 1.924 [1.04, 2.26] and HR 4.969 [1.00, 3.26], respectively). At the end of the follow-up period of 730 days in the entire cohort, 244 (3.86%) died, including 149 (43.7%) in the high and low-grade stages of AKI and 95 (1.6%) in the non-AKI group, while in the AKD group 186 (33.1%) patients and in the non-AKD group 58 (1.0%) patients died. The causes of death were complications/progression of brain tumor in 91 (37.2%) patients, stroke/intracerebral hemorrhage in 54 (22.1%), and myocardial infarction/heart failure in 99 (40.6%) patients. Mean survival time was longer in patients on propofol than on sevoflurane (729.6 (95% CI 728.2, 730.6) vs. 660.4 (95% CI 664.1, 665.4) days; p<0.001) (**S1A Fig**), patients without hypotension during the operation (708.8 (95% CI 688.4, 729.2) vs. 670.7 (95% CI 665.5, 675.2) days; p = 0.018) (**S1B Fig**) and in patients without the development of AKD (709.7 (95% CI 706.6, 712.5) vs. 527.7 (95% CI 510.3, 544.8) days; p<0.001) (**S1C Fig**).

## Discussion

The main finding of this study is that ICU patients, after neurosurgical procedures treated with balanced anesthesia with sevoflurane, more frequently developed AKI and AKD by the end of their ICU stay. Patients with prior comorbidities like diabetes and coronary disease were more prone to kidney injury, as well as patients with hypotension during surgical procedures. This is the first study which showed the impact on different anaesthetics, namely sevoflurane, on development of AKD.

**Table 2. Demographic parameters, comorbidities, and differences in clinical variables and laboratory parameters between high and low stage AKI and non-AKI patients.**

| Variables | High stage AKI (N = 22) | Low stage AKI (N = 319) | Non-AKI (N = 5976) | p |
|---|---|---|---|---|
| Age (years) | 58.7±3.3 | 56.6±3.2 | 55.3±3.0 | 0.49 |
| Sex (males) N(%) | 16 (72.7) | 210 (65.8) | 4251 (71.2) | 0.27 |
| BMI | 27.2±2.1 | 26.±2.2 | 27.1±2.0 | 0.77 |
| Comorbidities N(%) | | | | |
| *Diabetes* | 3 (13.6) | 53 (16.6) | 511 (8.5) | <0.001 |
| *Hypertension* | 5 (22.7) | 67 (21.0) | 1066 (17.8) | 0.69 |
| *Coronary heart disease* | 4 (18.2) | 66 (20.6) | 339 (5.7) | <0.001 |
| Indication for surgery | | | | |
| *Aneurism* | 12 (54.6) | 167 (52.3) | 3002 (50.2) | 0.89 |
| *Tumour* | 5 (22.7) | 89 (27.8) | 1195 (19.9) | 0.73 |
| *Subdural haematoma* | 5 (22.7) | 63 (19.7) | 1779 (29.7) | 0.22 |
| ASA score | 2.77±0.8 | 2.71±0.8 | 2.46±0.5 | 0.09 |
| GCS score | 10.8±1.2 | 11.3±1.7 | 11.8±1.9 | 0.17 |
| Duration of surgery (minutes) | 309.3±29.1 | 322.5±31.1 | 333.2±31.6 | 0.42 |
| Type of anaesthesia | | | | |
| *Propofol N(%)* | 13 (59.1) | 172 (53.9) | 4493 (75.2) | <0.001 |
| *Sevoflurane N(%)* | 9 (40.9) | 138 (46.1) | 1483 (24.8) | <0.001 |
| Anaesthesia induction | 2.1±0.5 | 2.2±0.4 | 2.2±0.5 | 0.84 |
| *Propofol dose (mg/kg)* | 0.22 (0.11–0.30) | 0.20 (0.10–0.25) | 0.19 (0.09–0.24) | 0.28 |
| *Opioid dose (mg/kg)* | 0.60±0.1 | 0.61±0.1 | 0.60±0.1 | 0.79 |
| *Muscle relaxant dose (mg/kg)* | 0.11 (0.10–0.15) | 0.12 (0.10–0.16) | 0.12 (0.10–0.15) | 0.67 |
| Intravenous anaesthesia | 0.24 (0.12–0.32) | 0.22 (0.11–0.30) | 0.23 (0.12–0.31) | 0.81 |
| *Propofol dose (mg/kg/min)* | 0.11 (0.09–0.12) | 0.10 (0.08–0.11) | 0.11 (0.08–0.13) | 0.75 |
| *Opioid dose (mg(kg/h)* | 2.2±0.5 | 2.0±0.5 | 1.8±0.3 | 0.02 |
| *Muscle relaxant dose (bolus mg/kg)* | | | | |
| *MAC dose (vol%)* | | | | |
| Days in ICU | 36.5±3.5 | 24.2±2.0 | 12.1±1.3 | 0.02 |
| Hemodynamic status | | | | |
| Surgery: | | | | |
| *Hypotension during operation Yes N(%)* | 5 (22.7) | 60 (18.8) | 752 (12.5) | <0.01 |
| *Mean arterial pressure (mmHg)* | 64.1±8.1 | 63.8±7.9 | 63.5±7.6 | 0.71 |
| *Mean administration of fluids (mL/h)* | 618±42.6 | 595±41.2 | 621±44.4 | 0.43 |
| *Hemorrhage (mL)* | 146±17.4 | 154±18.2 | 158±19.2 | 0.11 |
| Systolic blood pressure (mmHg) baseline | 149.4±17.5 | 144.2±17.0 | 142.6±16.9 | 0.18 |
| Systolic blood pressure (mmHg) after | 135.6±16.1 | 134.8±15.8 | 139.2±16.1 | 0.61 |
| Diastolic blood pressure (mmHg) baseline | 68.7±10.1 | 70.2±10.7 | 74.6±11.2 | 0.20 |
| Diastolic blood pressure (mmHg) after | 65.6±10.0 | 70.2±10.5 | 72.±10.8 | 0.06 |
| Mean arterial pressure (mmHg) baseline | 83.4±11.2 | 81.±10.5 | 81.2±10.1 | 0.34 |
| Mean arterial pressure (mmHg) after | 82.2±10.9 | 80.4±10.1 | 81.8±10.7 | 0.43 |
| CVP (mmHg) baseline | 12 (9–15) | 11 (8–14) | 10 (7–14) | 0.19 |
| CVP (mmHg) after | 7 (4–10) | 7 (5–10) | 8 (5–10) | 0.24 |
| Urinary output (ml/h) after | 25 (10–42) | 60 (31–89) | 92 (74–118) | 0.04 |
| Laboratory data | | | | |
| Sodium (mmol/l) baseline | 137 (134–141) | 135 (131–139) | 136 (132–140) | 0.51 |
| Sodium (mmol/l) after | 136 (133–140) | 135 (132–139) | 136 (132–140) | 0.86 |
| WBC (x10$^9$/l) baseline | 11.7±1.9 | 11.4±1.7 | 11.1±1.5 | 0.73 |
| WBC (x10$^9$/l) after | 10.5±0.9 | 10.4±0.9 | 10.2±0.8 | 0.88 |
| Platelets (x10$^9$/l) baseline | 186 (154–214) | 172 (140–208) | 204 (168–239) | 0.29 |
| Platelets (x10$^9$/l) after | 179 (141–212) | 173 (141–209) | 191 (160–207) | 0.41 |
| Creatinine (umol/l) baseline | 83.6±4.2 | 81.1±4.1 | 79.±4.1 | 0.65 |
| Creatinine (umol/l) after | 596.2±22.3 | 171.2±9.3 | 81.8±4.0 | <0.001 |

*(Continued)*

**Table 2.** (Continued)

| Variables | High stage AKI (N = 22) | Low stage AKI (N = 319) | Non-AKI (N = 5976) | p |
|---|---|---|---|---|
| GFR (ml/min/1.73m$^2$) baseline | 81 (41–119) | 83 (43–120) | 85 (45–122) | 0.56 |
| GFR (ml/min/1.73m$^2$) after | 8 (5–13) | 38 (17–52) | 87 (48–124) | <0.001 |
| BUN (umol/l) baseline | 9.2±1.8 | 8.8±1.8 | 8.1±1.5 | 0.13 |
| BUN (umol/l) after | 57.4±9.1 | 14.3±4.2 | 7.7±2.0 | <0.001 |
| C-reactive protein (mg/l) baseline | 24 (8–41) | 32 (10–49) | 22 (7–39) | 0.77 |
| C-reactive protein (mg/l) after | 17 (4–29) | 13 (3–23) | 15 (4–25) | 0.82 |
| Survival days | 477 (219–677) | 523 (301–642) | 590 (342–717) | 0.22 |

AKI-acute kidney injury; BMI-body mass index; ASA-American Society of Anesthesiologists; GCS-Glasgow coma score; MAC-minimal alveolar concentration; ICU-intensive care unit; CVP-central venous pressure; WBC-white blood count; GFR-estimated glomerular filtration rate; BUN-blood urea nitrogen; results are shown as mean +/- SD or median (interquartile range).

There are more and more reports that volatile anesthetics like sevoflurane powerfully modulate inflammation and, specifically, kidney injury caused by ischemia and low perfusion [18]. Moreover, its administration can be protective in pretreatment to prolonged ischemia like liver and brain and so called anesthetic preconditioning [19,20]. When administered after ischaemic insult, volatile anesthetics can be considered a new potential drug in protecting

**Table 3.** Unadjusted associations between covariates and development of AKI and/or AKD.

| AKI | | | |
|---|---|---|---|
| Variables | Odds Ratio | 95% CI | p |
| Age | 0.97 | 0.97–1.02 | 0.26 |
| Sex (Males) | 0.98 | 0.56–0.82 | 0.49 |
| Diabetes Yes | 3.54 | 0.25–49.08 | 0.35 |
| Coronary Disease Yes | 1.35 | 0.10–18.21 | 0.82 |
| Days in ICU | 0.95 | 0.91–0.99 | 0.33 |
| Duration of operation | 1.00 | 0.99–1.00 | 0.44 |
| Sevoflurane | 5.09 | 0.02–0.77 | 0.02 |
| Systolic blood pressure (mmHg) after | 0.77 | 0.42–0.66 | 0.89 |
| Diastolic blood pressure (mmHg) after | 1.82 | 1.00–1.32 | 0.19 |
| Hypotension during operation Yes | 3.43 | 0.87–1.80 | 0.06 |
| AKD | | | |
| Variables | Odds Ratio | 95% CI | p |
| Age | 0.89 | 0.94–1.03 | 0.59 |
| Sex (Males) | 0.66 | 0.41–0.77 | 0.34 |
| Diabetes Yes | 1.34 | 0.89–2.14 | 0.27 |
| Coronary Disease Yes | 5.98 | 0.41–6.60 | 0.19 |
| Days in ICU | 1.01 | 0.98–1.03 | 0.37 |
| Duration of operation | 1.00 | 0.99–1.00 | 0.60 |
| Sevoflurane | 4.98 | 0.79–1.15 | 0.04 |
| Systolic blood pressure (mmHg) after | 0.98 | 0.78–0.86 | 0.92 |
| Diastolic blood pressure (mmHg) after | 1.56 | 1.02–1.08 | 0.21 |
| Hypotension during operation Yes | 2.30 | 0.88–1.34 | 0.43 |

ICU-intensive care unit; GFR-estimated glomerular filtration rate; AKI-acute kidney injury; AKD-acute kidney disease.

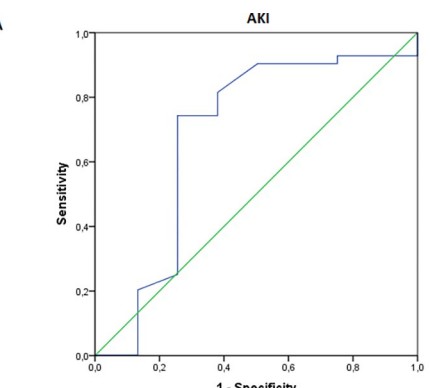
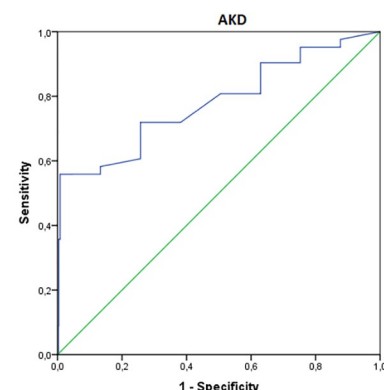

Fig 2. A and B. The receiver operating characteristics curves show the predictability of sevoflurane for (A) AKI and (B) AKD. AKI-acute kidney injury; AKD-acute kidney disease.

organ inflammation and ischaemia like in the kidney and heart [21]. Unfortunately, the reports of sevoflurane's toxic effects on renal function are still rather confusing. Ebert et al. [22,23] showed that prolonged sevoflurane administration to healthy volunteers did not result in clinically significant changes in biochemical markers of kidney or hepatic dysfunction. Even a meta-analysis by Ong Sio et al. [24] concluded that sevoflurane does not affect kidney function in apparently healthy adults without coexisting kidney disorders.

Contrary to these reports, studies showed that exposing healthy volunteers to sevoflurane for 8 hours and low fresh gas flow rates caused transient abnormalities in renal tubular injury biomarkers, e.g., urinary albumin, glucose, and alpha-glutathione-S-transferase [25]. The meta-analysis by Franzen et al. [26] revealed that using propofol is associated with a lower incidence of postoperative AKI compared with using volatile anesthetics, which we have confirmed with our results.

Sevoflurane has many effects on kidney function [27,28]. Urine output reduction and the increase in postoperative use of diuretics in different surgery types have been described compared to propofol anesthesia [12,29,30]. It is also reported to compromise renal perfusion and oxygenation in animals [31]. The results from two randomized control studies in adults and children showed the development of acute impairment of renal function, the reduction of urine output and sodium excretion, and the increase in plasma renin by sevoflurane [32,33]. Recent studies that explored the effects of neurogenic modulatory effects of sevoflurane on kidneys showed the increase in renal sympathetic activity as a mechanism for renal function impairment through vasoconstriction causing renal hypoperfusion, decrease in renal oxygenation and reductions in GFR [31,33,34]. Similar to our results, using volatile anesthetics like sevoflurane is reported to be associated with an increased risk and severity of AKI compared to propofol [12,26,35,36]. Patients on sevoflurane had an increased OR of 5.09 for AKI development.

The potential reasons for these observations include 1) toxic neurogenic modulatory effects of sevoflurane on kidneys, 2) hypotension during the operation was more common among patients with high and low AKI stages. Additional factors like bleeding and hypotension can further increase this risk for AKI [37]. Recent studies confirmed hypotension during operation is an independent risk factor for AKI [38]. In addition, ischemia and hypoxia as a direct result of hypotension can lead to AKI development [39].

**Table 4. Demographic parameters, comorbidities, and differences in clinical variables and laboratory parameters between AKD and non-AKD patients.**

| Variables | AKD (N = 562) | Non-AKD (N = 5755) | p |
|---|---|---|---|
| Age (years) | 62.0±3.9 | 53.8±3.1 | 0.08 |
| Sex (males) N(%) | 412 (73.3) | 4023 (69.9) | 0.73 |
| BMI | 27.3±2.3 | 27.0±2.1 | 0.93 |
| Comorbidities N(%) | | | |
| *Diabetes* | 123 (21.8) | 444 (7.7) | <0.001 |
| *Hypertension* | 129 (22.9) | 1009 (17.5) | 0.29 |
| *Coronary heart disease* | 111 (19.7) | 298 (5.2) | <0.001 |
| Indication for surgery | | | |
| *Aneurism* | 287 (51.1) | 2894 (50.3) | 0.73 |
| *Tumour* | 126 (22.4) | 1163 (20.2) | 0.61 |
| *Subdural haematoma* | 149 (26.5) | 1698 (29.5) | 0.83 |
| ASA score | 2.84±0.9 | 2.65±0.6 | 0.29 |
| GCS score | 11.1±1.2 | 11.6±1.7 | 0.46 |
| Duration of surgery (minutes) | 303.3±27.2 | 334.4±32.0 | 0.29 |
| Type of anaesthesia | | | |
| *Propofol* | 307 (54.6) | 4371 (75.9) | <0.001 |
| *Sevoflurane* | 255 (45.4) | 1384 (24.1) | <0.001 |
| Anaesthesia induction | 2.1±0.4 | 2.2±0.4 | 0.82 |
| *Propofol dose (mg/kg)* | 0.22 (0.11–0.31) | 0.21 (0.10–0.27) | 0.49 |
| *Opioid dose (mg/kg)* | 0.61±0.1 | 0.61±0.1 | 0.90 |
| *Muscle relaxant dose (mg/kg)* | 0.12 (0.11–0.16) | 0.11 (0.10–0.15) | 0.53 |
| Intravenous anaesthesia | 0.23 (0.12–0.30) | 0.22 (0.10–0.29) | 0.82 |
| *Propofol dose (mg/kg/min)* | 0.11 (0.09–0.12) | 0.11 (0.08–0.12) | 0.93 |
| *Opioid dose (mg(kg/h)* | 2.1±0.5 | 2.0±0.5 | 0.33 |
| *Muscle relaxant dose (bolus mg/kg)* | | | |
| *MAC dose (vol%)* | | | |
| AKI Yes N(%) | 175 (31.1) | 166 (2.9) | <0.001 |
| Hemodynamic status | | | |
| Surgery: | | | |
| *Hypotension during operation Yes N(%)* | 137 (24.2) | 680 (11.8) | <0.01 |
| *Mean arterial pressure (mmHg)* | 64.0±8.2 | 63.±8.0 | 0.88 |
| *Mean administration of fluids (mL/h)* | 614±42.0 | 59±41.4 | 0.54 |
| *Hemorrhage (mL)* | 14±17.9 | 152±18.1 | 0.32 |
| Systolic blood pressure (mmHg) baseline | 148.3±17.1 | 143.2±16.5 | 0.36 |
| Systolic blood pressure (mmHg) after | 131.2±15.5 | 141.6±16.4 | 0.19 |
| Diastolic blood pressure (mmHg) baseline | 67.3±9.9 | 72.5±10.7 | 0.33 |
| Diastolic blood pressure (mmHg) after | 63.4±9.2 | 73.6±10.9 | <0.01 |
| Mean arterial pressure (mmHg) baseline | 81.8±10.3 | 82.8±10.9 | 0.48 |
| Mean arterial pressure (mmHg) after | 63.1±8.2 | 82.2±10.3 | <0.01 |
| CVP (mmHg) baseline | 12 (9–14) | 11 (7–14) | 0.27 |
| CVP (mmHg) after | 8 (5–11) | 8 (5–12) | 0.69 |
| Urinary output (ml/h) after | 42 (21–58) | 100 (79–127) | <0.001 |
| Laboratory data | | | |
| Sodium (mmol/l) baseline | 137 (134–141) | 135 (130–139) | 0.42 |
| Sodium (mmol/l) after | 137 (133–142) | 134 (129–138) | 0.22 |
| WBC (x10$^9$/l) baseline | 11.2±1.7 | 11.8±1.9 | 0.69 |
| WBC (x10$^9$/l) after | 9.7±0.4 | 12.3±2.3 | 0.12 |
| Platelets (x10$^9$/l) baseline | 182 (154–207) | 175 (151–201) | 0.47 |
| Platelets (x10$^9$/l) after | 180 (149–207) | 199 (167–228) | 0.29 |
| Creatinine (umol/l) baseline | 84.1±4.9 | 80.5±3.7 | 0.39 |
| Creatinine (umol/l) after | 418.2±19.5 | 132±7.1 | <0.001 |

(*Continued*)

**Table 4.** (Continued)

| Variables | AKD (N = 562) | Non-AKD (N = 5755) | p |
|---|---|---|---|
| GFR (ml/min/1.73m$^2$) baseline | 80 (41–118) | 84 (45–121) | 0.42 |
| GFR (ml/min/1.73m$^2$) after | 12 (7–18) | 52 (28–77) | <0.001 |
| BUN (umol/l) baseline | 9.5±1.8 | 8.2±1.7 | 0.19 |
| BUN (umol/l) after | 32.5±6.2 | 10.2±4.2 | <0.001 |
| C-reactive protein (mg/l) baseline | 26 (10–45) | 30 (9–46) | 0.84 |
| C-reactive protein (mg/l) after | 15 (4–21) | 15 (5–24) | 0.99 |
| Survival days | 288 (92–437) | 694 (559–722) | <0.001 |

AKD-acute kidney disease; AKI-acute kidney injury; BMI-body mass index; ASA-American Society of Anesthesiologists; GCS-Glasgow coma score; MAC-minimal alveolar concentration; ICU-intensive care unit; CVP-central venous pressure; WBC-white blood count; GFR-estimated glomerular filtration rate; BUN-blood urea nitrogen; results are shown as mean +/- SD or median (interquartile range).

We found a higher incidence of hypertension than other comorbidities in all subgroups. This may indicate higher vulnerability to hypotension and AKI development, also noted in a study by Hallqvist et al. [40].

Diabetes and prior coronary disease are well-recognized risk factors for postoperative AKI development [41]. In our study, there was a significantly higher number of patients with diabetes and coronary disease in patients with high and low AKI stages.

The studies that reported kidney toxicity caused by sevoflurane suggested its toxicity under specific circumstances [12,27–29,42]. Our results indicate prior comorbidities associated with increased risk for AKI like diabetes, coronary disease, and hypotension during operation, which can increase the risk of drug toxicity. This may motivate choosing propofol-based anesthesia in patients with increased risk of postoperative AKI due to preexisting high risk of renal ischemia.

The data on the AKI incidence in neurosurgery ICU are relatively scarce and are primarily based on specific cohorts of patients [43]. The incidence of moderate to severe AKI was low in our cohort probably due to the exclusion of patients with CKD, who are more prone to develop AKI episodes, as in other studies which analyzed the incidence of AKI after neurosurgery (0.5% vs. 1.3%; 2.4%; 1.0%) [4,9,10], but interestingly, none of these studies analyzed or described the usage of different anesthetics. The impact of AKI after the neurosurgical procedure was independently associated with shorter survival during ICU stay in our research and other papers that analyzed the incidence of AKI after neurosurgery [4,9,10]. AKI is closely associated with an increased risk of developing CKD, depending on the AKI severity and other comorbidities [44]. The duration of AKI is also independently associated with a high CKD risk [45]. Different comorbidities like diabetes double the risk of AKI and AKD [46] similar to our cohort. These comorbidities, joined with the impacts of renal hypoperfusion caused by intraoperative hypotension and toxic kidney effects of sevoflurane, are most probably responsible for AKD incidence of 8.9% at the end of ICU stay in our study. Based on our results, patients who developed AKI and later AKD had significantly shorter survival than patients without AKI and AKD, similar to the meta-analysis from Coca et al. [15]. AKD had even a stronger association with higher mortality than AKI development in our group of patients with HR of 4.96. Nevertheless, sevoflurane and hypotension during operation as probable culprits for AKI and AKD development remained two variables, except that AKD was associated with higher mortality. Our results on the association of hypotension during operation and a history of other comorbidities, such as diabetes mellitus and coronary artery disease, with the risk of AKI and AKD development may be generalizable to other non-neurosurgical postoperative settings where the administration of sevoflurane should be avoided.

Our study has some limitations. First, the study was retrospectively performed in a single center. Future trials should be designed to identify high-risk patients for AKI and AKD development prospectively. Second, we used serum creatinine, BUN, and urine output to determine AKI with RIFLE criteria. While some may view the reliance on RIFLE as a limitation, it is noteworthy that research comparing the performance of AKIN, KDIGO, and RIFLE criteria has shown that RIFLE can effectively identify a higher percentage of patients in the early stages of AKI [47].

Additionally, studies have indicated that the RIFLE criteria possess excellent discriminatory power for predicting mortality compared to the other two definitions. This suggests that, despite being an older classification system, RIFLE may still provide valuable insights in certain clinical contexts, particularly in identifying patients at risk for adverse outcomes early in the course of AKI.

Ultimately, the choice of criteria may depend on the specific objectives of a study or clinical assessment, and it is essential to consider the strengths and limitations of each classification system when interpreting results. The addition of 24-h proteinuria and/or urine glucose excretion could provide a better diagnostic tool in assessing the severity of kidney injury, so our results should be interpreted with caution. Furthermore, we did not analyze AKI biomarkers, e.g., Cystatin C, which could provide earlier detection of subclinical AKI. Third, we did not have measured tubular injury biomarkers. In future studies, the availability of these biomarkers could delineate the intensity and location of injury following injurious exposures, such as sevoflurane nephrotoxicity. Fourth, we did not collect data during a follow-up period regarding the stage of CKD or progression to end-stage kidney disease, which could add more significance to our findings on association of sevoflurane and its long-term effects on kidney function. Fifth, our results could not confirm a causal relationship between sevoflurane on renal function. Approximately one-quarter of the initial cohort of patients undergoing neurosurgical procedures did not have an available postoperative creatinine value. Although this data does not differ significantly from the literature [48], this significant proportion of excluded patients and the exclusion of patients with CKD does not allow us to have real data on the incidence of AKI and AKD in this setting.

## Conclusions

ICU patients treated with balanced anesthesia with sevoflurane for the neurosurgical procedures had more frequent AKI and AKD at the end of ICU stay and had shorter survival than patients without AKI and AKD. Based on our results, we can presume that renal toxicity of sevoflurane is pronounced in special conditions like hypotension during operation and with a history of other comorbidities like diabetes and coronary disease.

## Supporting information

**S1 Fig.** A-C. Kaplan-Meier analysis of survival probability according to AKD versus non-AKD (A), presence of intra-operative hypotension (B) and according to propofol versus sevoflurane (C).
(TIF)

## Author Contributions

**Conceptualization:** Vedran Premuzic, Nikolina Basic-Jukic.

**Data curation:** Marin Lozic, Josip Kovacevic, Vladimir Prelevic, Marina Peklic.

**Formal analysis:** Vasilije Stambolija, Marin Lozic, Josip Kovacevic, Vladimir Prelevic, Marina Peklic, Ante Sekulic.

**Investigation:** Marin Lozic, Josip Kovacevic, Vladimir Prelevic, Marina Peklic, Miroslav Scap, Ante Sekulic.

**Methodology:** Vasilije Stambolija, Marin Lozic, Josip Kovacevic, Vladimir Prelevic, Marina Peklic, Miroslav Scap, Ante Sekulic.

**Project administration:** Vedran Premuzic.

**Supervision:** Vedran Premuzic, Miroslav Scap, Nikolina Basic-Jukic, Slobodan Mihaljevic, Kianoush B. Kashani.

**Validation:** Vedran Premuzic, Vasilije Stambolija, Nikolina Basic-Jukic, Slobodan Mihaljevic.

**Visualization:** Vedran Premuzic, Vasilije Stambolija, Nikolina Basic-Jukic, Slobodan Mihaljevic, Kianoush B. Kashani.

**Writing – original draft:** Vedran Premuzic, Kianoush B. Kashani.

**Writing – review & editing:** Vedran Premuzic, Kianoush B. Kashani.

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
