## [Decision Letter · Decision Letter 0]

7 Aug 2024

PONE-D-24-18913Risk factors for AKI and persistent AKD after neurosurgical procedurePLOS ONE

Dear Dr. Premuzic,

Thank you for submitting your manuscript to PLOS ONE. After careful consideration, we feel that it has merit but does not fully meet PLOS ONE’s publication criteria as it currently stands. Therefore, we invite you to submit a revised version of the manuscript that addresses the points raised during the review process.

We look forward to receiving your revised manuscript.

Kind regards,

Stefano Turi

Academic Editor

PLOS ONE

3.Please include captions for your Supporting Information files at the end of your manuscript, and update any in-text citations to match accordingly. Please see our Supporting Information guidelines for more information: http://journals.plos.org/plosone/s/supporting-information. 

Reviewers' comments:

Reviewer's Responses to Questions

**Comments to the Author**

1. Is the manuscript technically sound, and do the data support the conclusions?

Reviewer #1: Yes

Reviewer #2: Partly

Reviewer #3: Partly

2. Has the statistical analysis been performed appropriately and rigorously? 

Reviewer #1: Yes

Reviewer #2: No

Reviewer #3: Yes

3. Have the authors made all data underlying the findings in their manuscript fully available?

Reviewer #1: Yes

Reviewer #2: Yes

Reviewer #3: No

4. Is the manuscript presented in an intelligible fashion and written in standard English?

Reviewer #1: Yes

Reviewer #2: Yes

Reviewer #3: Yes

5. Review Comments to the Author

Reviewer #1: Abstract should start with a couple of sentences for introducing the topic. For example, the rate of AKI described in this specific setting. And that AKD is a recent nosological entity poorly studied after neurosurgery.

Introduction. “…risk factors for AKI are … use of mannitol… and elevated postoperative cystatin C…”. These two aspects are not correct. (1) mannitol is not nephrotoxic, consequently you should write something like “dehydration consequent to intraoperative mannitol use; (2) cystatin C is a AKI marker and not a risk factor for AKI, and consequenlty should be removed.

Introduction. A sentence on the AKD as a recent nosological entity different from AKI should be included.

Methods. A definition of AKD should be included.

Methods. Authors excluded patients with CKD. So, please, include the adopted definition of CKD.

Methods. You should report what did you consider as the minimum acceptable number of available creatinine values during ICU stay in order to include a patient into your study. I presume that at least one or two postoperative creatinine values were considered as inclusion criterium. Correct? You should report in the flow chart how many patients you have excluded for not available postoperative creatinine values (which is an interesting information).

Results. You did not include the value (parients and percentage) of AKI incidence. Please correct.

By curiosity. Did you test any tubular biomarker in this setting? It would be very interesting a sub-analysis, even on a small cohort of patients, in order to speculate in which part of the tubule acts the sevoflurane nephrotoxicity.

Discussion. Reference 21, could you please specify which biomarkers?

Discussion. “The incidence of the high AKI stage in our cohort was low…” even because you excluded patients with CKD, who are more prone to develop AKI episodes. This aspect should be included.

Discussion. I would include a short paragraph on the potential generalizability of the results to other surgical/postoperative settings. Same anesthetic drugs? Same expected results?

Reviewer #2: Dear Authors,

I have read your work with interest. I believe some aspects need to be clarified further, and additional analyses should be included. Here are my observations:

Title: The title of the work does not reflect the primary objective, which is to evaluate the effect of different anesthetics on the incidence of AKI and AKD. Therefore, I think it should be modified accordingly.

Methods: How were patients with CKD excluded? How long before admission was the baseline creatinine measured?

Results: It is highlighted that both the use of sevoflurane and hypotension are associated with a high risk of AKI. However, I believe a multivariate analysis should be added to understand how these factors may interact with each other. The same applies to the risk of AKD and death.

Choice of anesthetics: The reasons for choosing different anesthetics are not clear to me. Are there clinical reasons, or are they based on the type of surgery? This should be explained to understand if any underlying conditions of the patients could have influenced the risk of AKI. For example, I did not understand if those who received sevoflurane experienced more hypotension. Could this be a cause of renal damage?

Study period: The study includes the enrollment of patients treated over approximately 10 years. There may have been changes in surgical techniques during this period that could influence the renal outcomes of patients (e.g., more or fewer hypotensive episodes).

Discussion and conclusions: Since the discussion and conclusions are focused on the use of different anesthetics, it might be helpful to add a comparison table of the patient population using propofol vs. sevoflurane, which should be discussed separately.

I believe each of these points should be reviewed and discussed to increase the clinical significance and impact of the paper.

Reviewer #3: The authors have investigated the outcome of AKI/AKD in association with anesthetic modality: propofol or sevoflurane. This is an important clinical problem and several studies have already showed that volatile anesthetics has a higher risk for AKI which was summarized in a recent meta-analysis. Interestingly, the authors do not mention this study. Also, the authors talk about hypoperfusion being the main cause of renal injury yet hypoperfusion is the subsequent effect of the actual underlying mechanism. These mechanisms should be more profoundly discussed. Even though the scope of the study provides clinical merit and important information, data is missing or not presented in an understandable way. I also struggle with the novelty of this study, as the literature (which is not cited in this manuscript) already supports the hypothesis that volatile agents increase the risk for AKI. More needs to be added and revised to provide more rigorous data, I have listed some other specific issues below.

- All readers are not renal scientists hence a definition between AKI and AKD needs to be clarified in the introduction. It also needs to be made more clear why AKI/AKD is a significant clinical problem.

- Why was RIFLE criteria used when KDIGO has updated the international standard on classifying AKI?

- The abstract is not clearly summarizing the manuscript. This needs to be revised.

- The authors are using abbreviations without defining them. i.e., RIFLE, BMI, ASA, CKD-MDRD.

- There are no clearly given inclusion and exclusion criteria in the methods section.

- Line 169-171: please provide this data as "all patients, the propofol group, and the sevoflurane group" so the reader understands if there were any differences between the anesthetic groups with prior comorbidities.

- Results: I find it quite difficult to follow the results. It is unfortunately written in a rather abrupt and discontinuing way so it gets hard to understand the actual data. Please revise and section the results more clearly. The study also has a confusing subgrouping. The hypothesis is to notice differences in anesthetic modality hence tabular data needs to be presented as in these groups separated, not the AKI/AKD subgrouping. The presentation of data needs to be revised in order to adress the hypothesis and make it easier for the reader to understand.

- Supplemental data: this should be main data in the manuscript. Testing anesthetic agents you need to provide an assessment on anesthetic depth and not just "standard regimen" statement. Provide data on actual administered propofol doses and MAC values for the sevoflurane group. Mean opioid and muscle relaxant should also be provided for each anesthetic group. In general, intraoperative parameters should be provided in a table in the main text. i.e., mean blood pressure during surgery, mean administration of fluids and hemorrhage, etc. The OR analysis needs to be stratified for these parameters if there is a significant difference between the groups. Demographics also needs to be presented as main data and not supplemental.

- As a scientist very involved in this specific area, I am quite astonished that there is a significant lack of references that already tested this hypothesis and actually support the data. The OR for AKI in volatile vs propofol anesthesia independent of surgery (PMID 36794753), RCTs that shows an acute impairment of renal function in adults and children by sevoflurane and not propofol (PMID 35279277, 35330795). And studies actually show that renal sympathetic nerve activity is the cause of this renal impairment (vasoconstriction causing hypoperfusion and reductions in GFR, 35330795, 30770052) which was summarized in detail in an extensive review (PMID 36404210). The authors should do a more thorough literature search and discuss the mechanisms of volatile effects more profoundly. Much is already known hence the scope of this study is more a "me too" rather than completely novel. The authors needs to be more clear in what the novelty of this study is and justify their hypothesis as much is already known.

6. PLOS authors have the option to publish the peer review history of their article (what does this mean?). If published, this will include your full peer review and any attached files.

Reviewer #1: **Yes: **Marco Allinovi

Reviewer #2: No

Reviewer #3: No

---

## [Author Response · Author response to Decision Letter 0]

17 Sep 2024

Zagreb, Croatia, September 16 2024

Dear Editor,

Please see the revised version of our manuscript (PONE-D-24-18913) entitled "The effect of different anesthetics on the incidence of AKI and AKD after neurosurgical procedure." Low LDL neuro ICU on We believe the revisions based on this peer-review exercise have significantly improved the quality of our manuscript. 

All authors have read and approved the submission of the manuscript; the manuscript has not been published and is not being considered for publication elsewhere, in whole or in part, in any language.

We hope you will find our study and obtained results interesting to be considered for publication in your prestige journal especially after changes which hopefully made the manuscript easier to comprehend. 

Reviewer #1:

Abstract should start with a couple of sentences for introducing the topic. For example, the rate of AKI described in this specific setting. And that AKD is a recent nosological entity poorly studied after neurosurgery.

As the reviewer suggested, we have added clarification regarding the AKI rate and AKD.

Lines 27-28: AKI Incidence after neurosurgical operations has been reported as 10-14%. The literature regarding the incidence of nosocomial acute kidney disease (AKD) following neurosurgery is scarce. 

Introduction. "…risk factors for AKI are … use of mannitol… and elevated postoperative cystatin C…". These two aspects are not correct. (1) mannitol is not nephrotoxic, consequently you should write something like "dehydration consequent to intraoperative mannitol use; (2) cystatin C is a AKI marker and not a risk factor for AKI, and consequenlty should be removed.

We would like to thank the reviewer for these important remarks. As suggested, we have rephrased the first sentence regarding mannitol use and removed the part with Cystatin C.

Line 60: ...blood loss, need for reoperation, dehydration or osmotic nephrosis following intraoperative mannitol administration, and...

Introduction. A sentence on the AKD as a recent nosological entity different from AKI should be included.

We have included a sentence on the AKD as suggested by the reviewer.

Lines 65-70: AKD is a recently recognized entity that is caused by kidney injury. It may follow an episode of AKI following maladaptive repair or occur without AKI if damage results in a progressive decline in kidney function after the 7th day of exposure. The development of AKD indicates changing renal function that could be strongly associated with long-term renal outcomes and major adverse kidney events (MAKEs) leading to CKD progression, ESKD, and mortality. 

Methods. A definition of AKD should be included.

We have included the definition of AKD as suggested.

Lines 122-123: AKD describes acute or subacute damage and/or loss of kidney function for a duration of between 7 and 90 days after exposure to an AKI-initiating event.

Methods. Authors excluded patients with CKD. So, please, include the adopted definition of CKD.

We have included the adopted definition of CKD in the revised version of the manuscript.

Lines 86-88: CKD was defined as kidney damage or glomerular filtration rate (GFR) <60 mL/min/1.73 m2 for at least three months, irrespective of cause (levey), and was adjudicated by reviewing medical records. 

Methods. You should report what did you consider as the minimum acceptable number of available creatinine values during ICU stay in order to include a patient into your study. I presume that at least one or two postoperative creatinine values were considered as inclusion criterium. Correct? You should report in the flow chart how many patients you have excluded for not available postoperative creatinine values (which is an interesting information).

The reviewer had assumed correctly. The availability of at least one postoperative creatinine values was considered an inclusion criterion, which we have added to the methods section of the revised version of the manuscript. We have already included the number of patients with incomplete or missing data during the ICU stay (N=3,816). In more than 90% of cases, patients were excluded for unavailable postoperative creatinine values. 

Lines 89-90: We have considered at least one postoperative creatinine values as inclusion criterium.

Results. You did not include the value (parients and percentage) of AKI incidence. Please correct.

We have included the AKI incidence (presented as percentages) in the results section of the revised manuscript version.

Line 217: AKI was diagnosed in 341 (5.39%) patients.

By curiosity. Did you test any tubular biomarker in this setting? It would be very interesting a sub-analysis, even on a small cohort of patients, in order to speculate in which part of the tubule acts the sevoflurane nephrotoxicity.

We completely agree with the reviewer's remark. Unfortunately, we did not have any measured injury biomarkers available. Nevertheless, we have included this in the limitations section of the manuscript.

Lines 367-369: Third, we did not have measured tubular injury biomarkers. In future studies, the availability of these biomarkers could delineate the intensity and location of injury following injurious exposures, such as sevoflurane nephrotoxicity. 

Discussion. Reference 21, could you please specify which biomarkers?

We have specified the specific markers as suggested by the reviewer.

Lines 305-307: Contrary to these reports, studies showed that exposing healthy volunteers to sevoflurane for 8 hours and low fresh gas flow rates caused transient abnormalities in renal tubular injury biomarkers, e.g., urinary albumin, glucose, and alpha-glutathione-S-transferase.

Discussion. "The incidence of the high AKI stage in our cohort was low…" even because you excluded patients with CKD, who are more prone to develop AKI episodes. This aspect should be included.

We have included the aspect of CKD in this sentence, as suggested by the reviewer.

Lines 340-341: The incidence of moderate to severe AKI was low in our cohort despite including patients with CKD, who are more prone to develop AKI episodes....

Discussion. I would include a short paragraph on the potential generalizability of the results to other surgical/postoperative settings. Same anesthetic drugs? Same expected results?

We agree with this remark, and we have included a short paragraph regarding the potential generalizability in the discussion section of the manuscript.

Lines 357-360: Our results on the association of hypotension during operation and a history of other comorbidities, such as diabetes mellitus and coronary artery disease, with the risk of AKI and AKD development may be generalizable to other non-neurosurgical postoperative settings where the administration of sevoflurane should be avoided.

Reviewer #2: 

Dear Authors,

I have read your work with interest. I believe some aspects need to be clarified further, and additional analyses should be included. Here are my observations:

Title: The title of the work does not reflect the primary objective, which is to evaluate the effect of different anesthetics on the incidence of AKI and AKD. Therefore, I think it should be modified accordingly.

We would like to thank the reviewer for this remark and hope that the revised title reflects the primary objective in more manner than before.

The effect of different anesthetics on the incidence of AKI and AKD after neurosurgical procedures 

Methods: How were patients with CKD excluded? How long before admission was the baseline creatinine measured?

We hope that we have explained the CKD exclusion criteria in more detail than in the original version of the manuscript.

Line 86-90: CKD was defined as kidney damage or glomerular filtration rate (GFR) <60 mL/min/1.73 m2 for at least three months, irrespective of cause, and was adjudicated by reviewing medical records. The baseline creatinine was measured routinely two weeks before admission. We have considered at least one postoperative creatinine values as inclusion criterium. 

Results: It is highlighted that both the use of sevoflurane and hypotension are associated with a high risk of AKI. However, I believe a multivariate analysis should be added to understand how these factors may interact with each other. The same applies to the risk of AKD and death.

We have added a multivariate analysis for both variables and possible interaction with AKI, AKD, and death. 

Lines 260-262: In another model, we analyzed the possible interaction between the use of sevoflurane and hypotension during the operation and the development of AKI and AKD. We did not find any statistically significant interaction for these two variables. 

Lines 271-273: In another model, we analyzed the possible interaction between the use of sevoflurane and hypotension during the operation and mortality. We did not find any statistically significant interaction for these variables. 

Choice of anesthetics: The reasons for choosing different anesthetics are not clear to me. Are there clinical reasons, or are they based on the type of surgery? This should be explained to understand if any underlying conditions of the patients could have influenced the risk of AKI. For example, I did not understand if those who received sevoflurane experienced more hypotension. Could this be a cause of renal damage?

We have explained the reasons for choosing different anesthetics in more detail, as suggested by the reviewer. In the new Table 1, we described the differences in demographic, clinical, and laboratory parameters between patients who received sevoflurane vs. propofol. There were no differences in hypotension between different anesthetics. 

Lines 100-103: The reasons for choosing different anesthetics were mostly based on the type of surgery. Propofol was used for patients with aneurysmal and arteriovenous vascular malformation, vascular surgery, and craniotomy for tumors. In contrast, sevoflurane was used for subdural hematoma or for urgent operations. 

Study period: The study includes the enrollment of patients treated over approximately 10 years. There may have been changes in surgical techniques during this period that could influence the renal outcomes of patients (e.g., more or fewer hypotensive episodes).

The neurosurgeons and the anaesthesiologists did not change significantly in either surgical techniques or anesthetic protocols during the study period, so we did not add this remark as a potential limitation to the revised version of the manuscript. 

Discussion and conclusions: Since the discussion and conclusions are focused on the use of different anesthetics, it might be helpful to add a comparison table of the patient population using propofol vs. sevoflurane, which should be discussed separately.

We agree with the reviewer entirely, so we have added a newly created Table 1. In this table, we described the differences in demographic, clinical, and laboratory parameters between patients who received sevoflurane vs. propofol. There were no differences in hypotension between different anesthetics.

Lines 199-204: We did not find differences in the number of patients who developed AKI, but there was a significantly higher number of patients who developed AKD in the sevoflurane group (16.9% vs. 6.3%). We did not find differences in age, sex, BMI, comorbidities, intraoperative parameters, or in administered doses of anesthetics, muscle relaxants, and opioids during anesthesia induction and during operation between these subgroups of patients. In logistic regression, we did not find a significant association between different variables for the development of AKI. 

Lines 313-320: The results from two randomized control studies in adults and children showed the development of acute impairment of renal function, the reduction of urine output and sodium excretion, and the increase in plasma renin by sevoflurane. Recent studies that explored the effects of neurogenic modulatory effects of sevoflurane on kidneys showed the increase in renal sympathetic activity as a mechanism for renal function impairment through vasoconstriction causing renal hypoperfusion, decrease in renal oxygenation and reductions in GFR. Similar to our results, using volatile anesthetics like sevoflurane is reported to be associated with an increased risk and severity of AKI compared to propofol

Lines 337-338: This may motivate choosing propofol-based anesthesia in patients with increased risk of postoperative AKI due to preexisting high risk of renal ischemia. 

Lines 357-360: Our results on the association of hypotension during operation and a history of other comorbidities, such as diabetes mellitus and coronary artery disease, with the risk of AKI and AKD development may be generalizable to other non-neurosurgical postoperative settings where the administration of sevoflurane should be avoided.

I believe each of these points should be reviewed and discussed to increase the clinical significance and impact of the paper.

Reviewer #3: 

The authors have investigated the outcome of AKI/AKD in association with anesthetic modality: propofol or sevoflurane. This is an important clinical problem and several studies have already showed that volatile anesthetics has a higher risk for AKI which was summarized in a recent meta-analysis. Interestingly, the authors do not mention this study. Also, the authors talk about hypoperfusion being the main cause of renal injury yet hypoperfusion is the subsequent effect of the actual underlying mechanism. These mechanisms should be more profoundly discussed. Even though the scope of the study provides clinical merit and important information, data is missing or not presented in an understandable way. I also struggle with the novelty of this study, as the literature (which is not cited in this manuscript) already supports the hypothesis that volatile agents increase the risk for AKI. More needs to be added and revised to provide more rigorous data, I have listed some other specific issues below.

- All readers are not renal scientists hence a definition between AKI and AKD needs to be clarified in the introduction. It also needs to be made more clear why AKI/AKD is a significant clinical problem.

We agree with the reviewer, and we added clarification on AKD and its significance in both the introduction and discussion sections of the revised version of the manuscript.

Lines 65-70: AKD is a recently recognized entity that is caused by kidney injury. It may follow an episode of AKI following maladaptive repair or occur without AKI if damage results in a progressive decline in kidney function after the 7th day of exposure. The development of AKD indicates changing renal function that could be strongly associated with long-term renal outcomes and major adverse kidney events (MAKEs) leading to CKD progression, ESKD, and mortality. 

Lines 122-124. AKD describes acute or subacute damage and/or loss of kidney function for a duration of between 7 and 90 days after exposure to an AKI-initiating event.

- Why was RIFLE criteria used when KDIGO has updated the international standard on classifying AKI?

The reviewer is correct. We have added KDIGO instead of RIFLE criteria.

Lines 110-112: Kidney Disease: Improving Global Outcomes (KDIGO) guidelines were used for the diagnosis of AKI by comparing changes in serum creatinine levels during hospitalization to baseline levels before admission to the hospital

- The abstract is not clearly summarizing the manuscript. This needs to be revised.

As suggested by the reviewer, the abstract has been revised and hopefully is more clearly summarizing the manuscript.

- The authors are using abbreviations without defining them. i.e., RIFLE, BMI, ASA, CKD-MDRD.

We apologize for the mistake and not defining all of the used abbreviations, which we have corrected in the revised version of the manuscript.

- There are no clearly given inclusion and exclusion criteria in the methods section.

We have added more detailed inclusion and exclusion criteria in the methods section, as suggested by the reviewer.

Lines 84-86: Patients were included if they were ≥18 years old and admitted to the hospital for neurosurgical procedures. Patients with a history of CKD at any stage and those with missing data d

---

## [Decision Letter · Decision Letter 1]

20 Oct 2024

PONE-D-24-18913R1The effect of different anesthetics on the incidence of AKI and AKD after neurosurgical proceduresPLOS ONE

Dear Dr. Premuzic,

Thank you for submitting your manuscript to PLOS ONE. After careful consideration, we feel that it has merit but does not fully meet PLOS ONE’s publication criteria as it currently stands. Therefore, we invite you to submit a revised version of the manuscript that addresses the points raised during the review process.

We look forward to receiving your revised manuscript.

Kind regards,

Stefano Turi

Academic Editor

PLOS ONE

Reviewers' comments:

Reviewer's Responses to Questions

**Comments to the Author**

1. If the authors have adequately addressed your comments raised in a previous round of review and you feel that this manuscript is now acceptable for publication, you may indicate that here to bypass the “Comments to the Author” section, enter your conflict of interest statement in the “Confidential to Editor” section, and submit your "Accept" recommendation.

Reviewer #1: All comments have been addressed

Reviewer #2: (No Response)

Reviewer #3: All comments have been addressed

2. Is the manuscript technically sound, and do the data support the conclusions?

Reviewer #1: Yes

Reviewer #2: Partly

Reviewer #3: Yes

3. Has the statistical analysis been performed appropriately and rigorously? 

Reviewer #1: Yes

Reviewer #2: No

Reviewer #3: Yes

4. Have the authors made all data underlying the findings in their manuscript fully available?

Reviewer #1: Yes

Reviewer #2: Yes

Reviewer #3: No

5. Is the manuscript presented in an intelligible fashion and written in standard English?

Reviewer #1: Yes

Reviewer #2: Yes

Reviewer #3: Yes

6. Review Comments to the Author

Reviewer #1: The Authors have worked hard to improve this article. Although of potential interest to Plos One readership, the manuscript still requires minor modifications before it can be considered for publication.

Abstract. AKI acronyms should be written. And further on in the abstract, acute kidney injury should be changed in AKI

Abstract. Please include the incidence of AKI and AKD in your cohort.

The following sentence should be rewritten:

"The incidence of moderate to severe AKI was low in our cohort despite including patients with CKD, who are more prone to develop AKI episodes"

correct sentence:

"The incidence of moderate to severe AKI was low in our cohort probably due to the exclusion of patients with CKD, who are more prone to develop AKI episodes"

I would include a sentence in your Limitations: Approximately one-quarter of the initial cohort of patients undergoing neurosurgical procedures did not have an available postoperative creatinine value. Although this data does not differ significantly from the literature [Villa G et al, 2021], this significant proportion of excluded patients and the exclusion of patients with CKD does not allow us to have real data on the incidence of AKI and AKD in this setting.

Villa G, De Rosa S, Scirè Calabrisotto C, Nerini A, Saitta T, Degl'Innocenti D, Paparella L, Bocciero V, Allinovi M, De Gaudio AR, Ostermann M, Romagnoli S. Perioperative use of serum creatinine and postoperative acute kidney injury: a single-centre, observational retrospective study to explore physicians' perception and practice. Perioper Med (Lond). 2021 May 25;10(1):13.

Reviewer #2: Dear Author,

I have reviewed the revised version of your paper. While I appreciate your efforts to respond to reviewers' comments, I have come other additional observations:

1. Please define AKD in the introduction.

2. There appears to be an inconsistency with the definition of AKI used in this paper. It seems that you have applied the RIFLE classification (including for defining AKI severity), whereas, in response to Reviewer 3’s critique, the text was modified to indicate the application of KDIGO criteria. It would be more accurate to clarify the specific AKI criteria actually employed and to discuss the rationale and potential limitations of this choice in the Discussion section.

3. It would be helpful to specify the type of statistical tests performed and to include the statistical values for the multivariate analyses added to the text (lines 260-262 and 271-273).

Reviewer #3: I have no further comments, the authors have made the necessary revisions. I appreciate the effort and the response from the authors.

7. PLOS authors have the option to publish the peer review history of their article (what does this mean?). If published, this will include your full peer review and any attached files.

Reviewer #1: No

Reviewer #2: No

Reviewer #3: No

---

## [Author Response · Author response to Decision Letter 1]

30 Oct 2024

Zagreb, Croatia, October 29 2024

Dear Editor,

Please see the revised version of our manuscript (PONE-D-24-18913R1) entitled "The effect of different anesthetics on the incidence of AKI and AKD after neurosurgical procedure." We believe the revisions based on this peer-review exercise have significantly improved the quality of our manuscript. 

All authors have read and approved the submission of the manuscript; the manuscript has not been published and is not being considered for publication elsewhere, in whole or in part, in any language.

We hope you will find our study and obtained results interesting to be considered for publication in your prestige journal especially after changes which hopefully made the manuscript easier to comprehend. 

Reviewer #1: The Authors have worked hard to improve this article. Although of potential interest to Plos One readership, the manuscript still requires minor modifications before it can be considered for publication.

Abstract. AKI acronyms should be written. And further on in the abstract, acute kidney injury should be changed in AKI

As suggested, we have changed acute kidney injury to AKI in the whole abstract.

Abstract. Please include the incidence of AKI and AKD in your cohort.

As suggested, we have included the incidence of AKI and AKD in the abstract.

The following sentence should be rewritten:

"The incidence of moderate to severe AKI was low in our cohort despite including patients with CKD, who are more prone to develop AKI episodes"

correct sentence:

"The incidence of moderate to severe AKI was low in our cohort probably due to the exclusion of patients with CKD, who are more prone to develop AKI episodes"

We have rewrite the sentence as suggested by the reviewer.

I would include a sentence in your Limitations: Approximately one-quarter of the initial cohort of patients undergoing neurosurgical procedures did not have an available postoperative creatinine value. Although this data does not differ significantly from the literature [Villa G et al, 2021], this significant proportion of excluded patients and the exclusion of patients with CKD does not allow us to have real data on the incidence of AKI and AKD in this setting.

Villa G, De Rosa S, Scirè Calabrisotto C, Nerini A, Saitta T, Degl'Innocenti D, Paparella L, Bocciero V, Allinovi M, De Gaudio AR, Ostermann M, Romagnoli S. Perioperative use of serum creatinine and postoperative acute kidney injury: a single-centre, observational retrospective study to explore physicians' perception and practice. Perioper Med (Lond). 2021 May 25;10(1):13.

We have included the proposed sentence in the limitations section of the manuscript and added the suggested reference.

Reviewer #2: Dear Author,

I have reviewed the revised version of your paper. While I appreciate your efforts to respond to reviewers' comments, I have come other additional observations:

1. Please define AKD in the introduction.

We have moved the definition of AKD from methods to the introduction section of the manuscript as suggested by the reviewer.

2. There appears to be an inconsistency with the definition of AKI used in this paper. It seems that you have applied the RIFLE classification (including for defining AKI severity), whereas, in response to Reviewer 3’s critique, the text was modified to indicate the application of KDIGO criteria. It would be more accurate to clarify the specific AKI criteria actually employed and to discuss the rationale and potential limitations of this choice in the Discussion section.

We agree with the reviewer. We have put back the exact criteria we have used for the definition of AKI and added the rationale and potential limitations of this choice in the limitations/discussion section of the manuscript.

Discussion: Lines: 363-372

While some may view the reliance on RIFLE as a limitation, it is noteworthy that research comparing the performance of AKIN, KDIGO, and RIFLE criteria has shown that RIFLE can effectively identify a higher percentage of patients in the early stages of AKI [44]. 

Additionally, studies have indicated that the RIFLE criteria possess excellent discriminatory power for predicting mortality compared to the other two definitions. This suggests that, despite being an older classification system, RIFLE may still provide valuable insights in certain clinical contexts, particularly in identifying patients at risk for adverse outcomes early in the course of AKI.

Ultimately, the choice of criteria may depend on the specific objectives of a study or clinical assessment, and it is essential to consider the strengths and limitations of each classification system when interpreting results. 

3. It would be helpful to specify the type of statistical tests performed and to include the statistical values for the multivariate analyses added to the text (lines 260-262 and 271-273).

We have specified the type of statistical tests and include statistical values for the multivariate analyses as suggested by the reviewer.

Reviewer #3: I have no further comments, the authors have made the necessary revisions. I appreciate the effort and the response from the authors.

Finally, we would like to thank all the reviewers and the editor for their critical appraisal, comments, and suggestions. Following several implemented changes, we hope that the current version of the manuscript can be considered for publication.

Sincerely

Vedran Premuzic

---

## [Decision Letter · Decision Letter 2]

25 Nov 2024

The effect of different anesthetics on the incidence of AKI and AKD after neurosurgical procedures

PONE-D-24-18913R2

Dear Dr. Premuzic,

We’re pleased to inform you that your manuscript has been judged scientifically suitable for publication and will be formally accepted for publication once it meets all outstanding technical requirements.

Kind regards,

Stefano Turi

Academic Editor

PLOS ONE

Reviewers' comments:

Reviewer's Responses to Questions

**Comments to the Author**

1. If the authors have adequately addressed your comments raised in a previous round of review and you feel that this manuscript is now acceptable for publication, you may indicate that here to bypass the “Comments to the Author” section, enter your conflict of interest statement in the “Confidential to Editor” section, and submit your "Accept" recommendation.

Reviewer #1: All comments have been addressed

Reviewer #2: All comments have been addressed

Reviewer #3: All comments have been addressed

2. Is the manuscript technically sound, and do the data support the conclusions?

Reviewer #1: Yes

Reviewer #2: Yes

Reviewer #3: Yes

3. Has the statistical analysis been performed appropriately and rigorously? 

Reviewer #1: Yes

Reviewer #2: Yes

Reviewer #3: Yes

4. Have the authors made all data underlying the findings in their manuscript fully available?

Reviewer #1: Yes

Reviewer #2: Yes

Reviewer #3: No

5. Is the manuscript presented in an intelligible fashion and written in standard English?

Reviewer #1: Yes

Reviewer #2: Yes

Reviewer #3: Yes

6. Review Comments to the Author

Reviewer #1: Although this manuscript has several limitations, the topic is interesting for nephrologists.

The Authors have worked hard to improve this research project, and they have met all criticisms raised by referees.

I feel the manuscript is now suitable for publication in Plos One.

Reviewer #2: Dear Authors, I think you are addressed my comments, and the paper has been improved.

I have no additional requests for you.

Reviewer #3: No further comments. The authors have adressed the reviewer issues and I appreciate the efforts they put in to this manuscript.

7. PLOS authors have the option to publish the peer review history of their article (what does this mean?). If published, this will include your full peer review and any attached files.

Reviewer #1: No

Reviewer #2: No

Reviewer #3: No

---

## [Editor Report · Acceptance letter]

7 Dec 2024

PONE-D-24-18913R2 

PLOS ONE

Dear Dr. Premuzic, 

I'm pleased to inform you that your manuscript has been deemed suitable for publication in PLOS ONE. Congratulations! Your manuscript is now being handed over to our production team.

Kind regards, 

on behalf of

Dr. Stefano Turi 

Academic Editor

PLOS ONE